# THREE WAYS THAT NON-DIFFERENTIABILITY AFFECTS NEURAL NETWORK TRAINING

## ABSTRACT

This paper investigates how non-differentiability affects three different aspects of the neural network training process. We first analyze fully connected neural networks with ReLU activations, for which we show that the continuously differentiable neural networks converge faster than non-differentiable neural networks. Next, we analyze the problem of $L_1$ regularization and show that the solutions produced by deep learning solvers are incorrect and counter-intuitive even for the $L_1$ penalized linear model. Finally, we analyze the Edge of Stability problem, where we show that all convex, non-smooth, Lipschitz continuous functions display unstable convergence, and provide an example of a result derived using twice differentiable functions which fails in the once differentiable setting. More generally, our results suggest that accounting for the non-linearity of neural networks in the training process is essential for us to develop better algorithms, and to get a better understanding of the training process in general.

## 1 INTRODUCTION

Gradient descent (GD) and its variants (Duchi et al. (2011); Kingma & Ba (2014); Lydia & Francis (2019); McMahan & Streeter (2010); Shi & Li (2021); Tieleman et al. (2012); Zhang (2018) to state a few) have played a pivotal role in driving the substantial progress achieved in image and language processing over the past two decades. These algorithms find widespread use in model training because they iteratively approach stationary points, where the loss function's gradient is zero (Bertsekas, 1997). Although GDs are designed for continuously differentiable loss functions, they are extensively used in the training of neural networks that lack differentiability. Remarkably, GD applied to non-differentiable neural networks (which we refer to as "Non-differentiable gradient descents" or NDGDs) exhibit asymptotic convergence to stationary points under mild regularity conditions (Bolte & Pauwels, 2021; Davis et al., 2020), mirroring the behavior of their differentiable counterparts.

The loss functions for non-differentiabile neural networks take a special form – they are often times differentiable almost everywhere. As a consequence, NDGDs seldom encounter non-differentiable points during the training process. This characteristic has led standard references in the field to assume that neural networks can be treated like differentiable functions, considering point discontinuities as inconsequential (see Section 6.1.5 of Stevens et al. (2020) for instance). This assumption has sparked renewed interest in studying the dynamics of gradient descents amidst complex loss landscapes (Ahn et al. (2022b); Dauphin et al. (2014); Reddi et al. (2019)), with fewer studies explicitly addressing non-differentiability Bertoin et al. (2021); Bolte & Pauwels (2021); Davis et al. (2020). Additionally, this assumption has led several theoretical papers to develop results for continuously differentiable functions, and then claim that their results hold for all cases by running simulations on non-differentiable functions (Experiment 6 in Ahn et al. (2022b), Section 3.1 in Ma et al. (2022), Section 6 in Zhang et al. (2022) to state a few).

In this paper, we show that this assumption is flawed, and analyze three cases where accounting for the non-differentiability dramatically alters the dynamics of the problem. Our contributions in the three cases are as follows:

- **Convergence analysis of ReLU networks** - We prove that the structural properties of NDGD and GD are not identical, and show that the convergence rates for NDGD sequences are much slower than GD sequences.

- **Solution to the LASSO problem** - We show that the solution to the LASSO problem is incorrect and counterintuitive even in the simplest case of the $L_1$ penalized linear model.
- **The Edge of Stability** - We show that the NDGD sequence for convex, non-differentiable, Lipschitz continuous functions never diverge to $\infty$ irrespective of the learning rate, and provide an example of a result developed for twice differentiable functions which fails in the once differentiable case.

In a broader context, our results show that treating NDGDs and GDs as equivalent significantly limits our understanding of the training process. Consequently, our results suggest that a way forward might be to treat the the dynamics of these two systems as distinct entities, rather than assuming their similarity, as is commonly done.

## 2 PRELIMINARIES

In the three sections that follow, we show how disregarding the non-differentiability of the loss function severely limits our understanding of different aspects of training dynamics. Each of these sections is largely self-contained, complete with a literature review, problem setup, and analysis. In this section, we cover the background and terminology which is common to all of these sections.

**NDGD v/s GD:** The crux of our analyses lies in the structural differences between GDs and NDGDs. For a continuously differentiable loss function, $f(\beta)$, the well-studied GD sequence (Bertsekas, 1997; Boyd et al., 2004) is described by the recursion

$$\beta_{t+1} = \beta_t - \alpha \nabla f(\beta_t), \tag{1}$$

where $\alpha$ is the learning rate, and $\beta_0$ is a randomly initialized starting point.

For a continuous non-differentiable loss function, $g(\beta)$, the NDGD recursion is given by

$$\tilde{\beta}_{t+1} = \tilde{\beta}_t - \alpha \widetilde{\nabla} g(\tilde{\beta}_t), \tag{2}$$

where $\widetilde{\nabla} g(\tilde{\beta}_t) = \nabla g(\tilde{\beta}_t)$ if the gradient exists at $\tilde{\beta}_t$, else it is a heuristic measure [1].

**Convex non-differentiable loss functions:** In this paper, we show the problems with this assumption using a variety of convex loss functions. The majority of our analysis is performed on the penalized single layer neural network regression problem with a non-positive response vector, which Kumar (2023) has shown to be convex. The loss function for this problem is given by

$$L_1(\beta; \mathbf{Z}, \mathbf{y}, \lambda_1, \lambda_2) = ||\mathbf{y} - \max(0, \mathbf{Z}\beta)||^2 + \lambda_1 ||\beta||_1 + \lambda_2 ||\beta||_2^2, \tag{3}$$

where $\beta$ is a $P \times 1$ parameter vector, $\mathbf{Z}$m is an $N \times P$ data matrix whose $i^{th}$ column is given by $z_i$, $\mathbf{y}$ is a $N \times 1$ non-positive response vector, $\lambda_1 \geq 0$ is the LASSO penalty, and $\lambda_2 \geq 0$ is the ridge penalty. Throughout this paper, we denote the learning rate by $\alpha$, and the $i^{th}$ entry in the parameter vector after $k$ iterations is $\beta_k[i]$, with $0 \leq i \leq P - 1$.

**The subgradient method:** Our analysis heavily relies on the observation that for a convex non-differentiable loss functions, the NDGD recursion described in (2) effectively implements the subgradient method (Bertsekas, 1997; Shor, 2012). The theoretical properties of this method are well studied in the optimization literature, with a concise overview of key findings available in the lecture notes of Boyd et al. (2003) and Tibshirani (2015). The subgradient method and GD recursions have different properties, and these differences play a key role in our subsequent discussions. We reference the relevant portions from the lecture notes as they arise.

**Constant, diminishing, and reducing step sizes:** Our formulations in equations (1) and (2) employ a constant learning rate. This choice bears some explaining, for which we first recall some definitions from the optimization literature (see Chapter 1 of Bertsekas (1997) for details). In the GD literature, the training regimen described in equation (1) is called a constant step-size regime. In this regime,

---

[1]https://rb.gy/g74av

the GD is initialized using the specified algorithm, and iterated with a constant learning rate until the loss stops decreasing. Although constant step-size GDs approach the stationary point, they rarely converge to it except in special cases.

To ensure provable convergence to the stationary point, theoretical studies often adopt the diminishing step-size regime. This regime is characterized by the iteration

$$\beta_{t+1} = \beta_t - \alpha_t \nabla f(\beta_t), \tag{4}$$

where $\lim_{t\to\infty} \alpha_t = 0$, and $\sum_t \alpha_t \to \infty$. Convergence results in this regime are derived under ideal conditions, and provide guarantees that the recursion will reach the stationary point at least in the theoretical limit.

In practice, diminishing step-size GDs are implemented with a stopping criteria which determines when the solution produced by the GD is good enough to end training. The combination of the diminishing step-size regime and stopping criteria commonly used in the neural network literature is what we term the "reducing" step-size regime. In this regime, training unfolds as follows: commencing with a weight of $\beta_0$, GD runs for some $N_0$ iterations with a learning rate of $\alpha_0$, followed by $N_1$ iterations with a reduced learning rate of $\alpha_1 < \alpha_0$, and so on, until the engineer is satisfied with the solution. Essentially, the reducing step-size regime involves chaining together a finite number (say $T$) of constant step-size GD runs, with the $(i+1)^{th}$ run having an initialization of $\beta_u$ and a learning rate of $\alpha_i$, where $u = \sum_{j=0}^{i-1} N_j$. In the reducing step size regime, the theoretical guarantees described in the diminishing step-size regime may not be realized. Nevertheless, the limiting dynamics of the sequence in this regime are exactly those of a constant step-size GD, with learning rates and weight initializations chosen corresponding to the values at the beginning of the $T^{th}$ run. Accordingly, in this paper, we exclusively study the constant step-size problem, assuming that the weights and learning rates have been appropriately chosen.

## 3 NDGD AND CONVERGENCE ANALYSIS

Recent theoretical work has shown that in the diminishing step-size regime, the NDGD sequence will converge to a local optimal solution under fairly general conditions. Unfortunately, these regularity conditions are rarely satisfied in practice. To illustrate this point, consider the Assumption A.3 in Davis et al. (2020), which requires that in the diminishing step size regime, the sum of the learning rates should diverge. As we previously stated, this is a standard assumption even in the thoeretical convergence analysis of GDs. However, the prevalent step-size reduction we observed in most training scripts involves decaying the learning rate by a constant multiplicative factor at predefined intervals (StepLR in PyTorch[2]). With this decay scheme, the learning rates form a convergent geometric series, thus violating the assumption.

For the differentiable case, we know that even when the learning rates form a convergent geometric series, the GD sequence will come within some vicinity of the optimal solution. Therefore, the question arises if the same holds true for NDGD sequences. More precisely, can we state that the dynamics of NDGD and GD sequences are identical?

**NDGD and GD dynamics are not identical:** The dynamics of the two sequences are not identical, and we demonstrate this by arriving at a contradiction. Suppose the sequence $\{\beta_k\}_{k=0}^{\infty}$ obtained by running NDGD on $L_2(\beta; \mathbf{Z}_1, \mathbf{q}_1) = L_1(\beta; \mathbf{Z}_1, \mathbf{q}_1, 0, 0.01)$ has the same properties as a GD sequence. Then, the entire NDGD sequence can be characterized using the "Capture Theorem" as described below

**Proposition 1.** *Let $x_{k+1} = x_k - \alpha \nabla f(x_k)$ be a sequence generated by running GD on a continuously differentiable function, $f(x)$ such that $f(x_{k+1}) \leq f(x_k)$. If $x^*$ is an isolated local minimum of $f(x)$ in a unit ball around $x^*$, and $||x_0 - x^*|| < 1$, then $||x_k - x^*|| < 1$ for all $k > 0$.*

*Proof.* Result follows by using $\bar{\epsilon} = 1$ in Proposition 1.2.5 of Bertsekas (1997). ∎

---

[2]https://rb.gy/pub4s

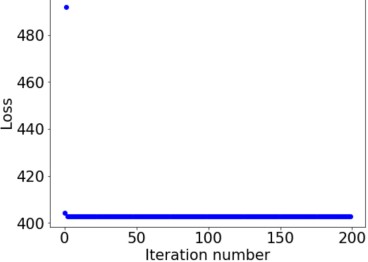 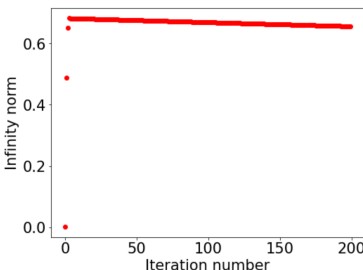

Figure 1: Plot of the loss function ($L_2(\beta_k; \mathbf{Z}_1, \mathbf{q}_1)$), and infinity norm ($||\beta_k||_\infty$) of the parameter vector as a function of the iteration number ($k$). Although the loss function is monotone decreasing, the infinity norm of the weights does not "capture", hence demonstrating that the dynamics of NDGD and GD are not identical

Therefore, if $||\beta_0||_\infty < 1$, and $L_2(\beta_{k+1}; \mathbf{Z}_1, \mathbf{q}_1) \leq L_2(\beta_k; \mathbf{Z}_1, \mathbf{q}_1)$ for all $k \geq 0$, then we expect $||\beta_k||_\infty < 1$ for all $k \geq 0$. To validate this claim, we first generate a $20 \times 500$ matrix $\mathbf{Z}_1$ with entries uniformly sampled from $[-1, 1]$. Next, we generate a $20 \times 1$ vector $\mathbf{q}_1$ with entries uniformly sampled on $[-1, 0]$. Finally, we generate the sequence $\{\beta_t\}_{t=0}^\infty$, by running NDGD on $L_2(\beta; \mathbf{Z}_1, \mathbf{q}_1)$ with the entries in $\beta_0$ uniformly sampled on $[-1, 1]$. As seen in Figure 1, although the loss function is a monotone decreasing function of the iteration number, the infinity norm of the NDGD sequence eventually takes values greater than 1, hence establishing the contradiction. The non-capturing described in Figure 1 is to be expected; as stated in Section 1 of Boyd et al. (2003), "Unlike the ordinary gradient method, the subgradient method is not a descent method; the function value can (and often does) increase".

**Convergence analysis and L-smoothness:** Given that the dynamics of GD and NDGD differ, the question arises what assumptions are appropriate for the study of convergence analysis in neural network training. Majority of the analyses in the literature (Défossez et al. (2020); Reddi et al. (2019); You et al. (2019); Zhang et al. (2022) to state a few) assume that the loss function, $f(\beta)$ is continuously differentiable, and is $L-smooth$, i.e., there exists a constant $L$, such that

$$||\nabla f(x) - \nabla f(y)|| \leq L||x - y|| \tag{5}$$

for all $x, y$ in the domain of $f$. However, this assumption breaks down for non-differentiable functions, where the gradient does not even exist at various points in the domain. This leads us to a pertinent inquiry: Can we salvage this assumption by confining its validity to a local context, possibly within a specific interval centered around the local optimal solution?

Unfortunately, even this cannot be done since ReLU networks typically have non-differentiable minima, except for trivial cases; as stated in Laurent & Brecht (2018), "local minima of ReLU networks are generically nondifferentiable. They cannot be waved away as a technicality, so any study of the loss surface of such network must invoke nonsmooth analysis". Moreover, assuming that the loss function is $L-smooth$ grossly over-estimates the speed of convergence of NDGDs. To illustrate this, consider that in order to achieve an error level of $\epsilon$, GD applied to an $L-smooth$ convex loss function, $f$, would require $O(1/\epsilon)$ steps for convergence. In contrast, NDGD applied to a convex non-differentiable loss function, $g$, satisfying the Lipschitz condition

$$||g(x) - g(y)|| \leq L||x - y|| \tag{6}$$

would take $O(1/\epsilon^2)$ steps (see Section 7.1.3 of Tibshirani (2015) for the result, and a visual illustration of how much slower $O(1/\epsilon^2)$ is than $O(1/\epsilon)$). Thus, it is likely that the vast literature on the convergence analysis of GDs for loss functions satisfying (5) likely have little bearing on the dynamics of convergence in modern non-smooth neural networks. More generally, we expect the speed of convergence for differentiable neural networks to be much faster than that of non-differentiable neural networks.

**Experiments with NDGD convergence:** To validate our intuition in a more general setting, we run experiments on different deep architectures described in Cohen et al. (2021). Each experiment entails

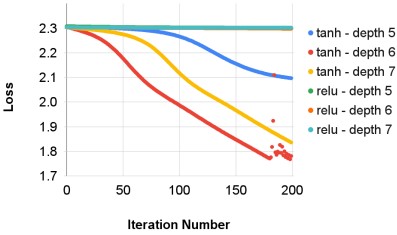
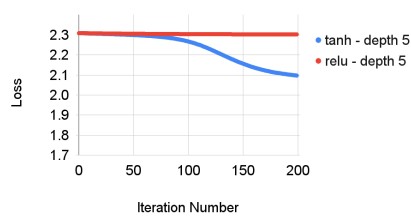

(a) Fully connected neural networks    (b) Convolutional neural networks

Figure 2: Comparison of the speeds of convergence for differentiable and non-differentiable neural network architectures. In every cases, we observe that convergence is faster on the differentiable architecture. We have verified in each case that the ReLU training has not stalled completely, but is slow progressing.

executing two sets of simulations that are identical in all aspects except for the choice of activation functions: one utilizes the differentiable tanh function, while the other employs the non-differentiable ReLU function. All simulations are conducted on an identical subset of 2000 images from CIFAR-10, utilizing the supplementary code in the above mentioned paper [3]; architecture details can be found in the appendix. Our hypothesis is that differentiable neural networks ought to converge significantly faster than non-differentiable neural networks. As illustrated in Figure 2, this hypothesis is borne out true for fully connected neural networks regardless of their depth (Figure 2a), as well as for more intricate architectures like convolutional neural networks (Figure 2b).

**The way forward:**    In this section, we show that NDGDs lacks the same convergence speed as GDs. Therefore, one possiblity to speed up the training process might be to focus on differentiable neural network architectures. Research into differentiable neural networks are in its early days (Agrawal et al., 2019; Amos & Kolter, 2017; Miconi, 2016), but work on this subject could provide valuable insights into we can speed up training.

## 4    NDGD AND THE LASSO PENALTY

Zero weights play a critical role in model compression (Blalock et al., 2020; Han et al., 2015a;b; Li et al., 2016), and it is desirable to force weights down to zero whenever we can do so without sacrificing too much accuracy. A common approach to induce sparsity is to penalize the $L_1$ norm of the model weights, with the penalizing factor commonly referred to as the LASSO penalty. In the statistics literature, loss functions involving penalized $L_1$ norms are typically minimized using specialized algorithms designed for different use cases (Efron et al., 2004; Friedman et al., 2008; Mazumder & Hastie, 2012; Tibshirani, 1996; Tibshirani et al., 2005; Zou & Hastie, 2005). However when training neural networks, the penalized $L_1$ norm is used with arbitrary non-convex functions, with the understanding that running NDGD on the problem results in a near sparse optimal solution (Bengio, 2012; Goodfellow et al., 2016). As stated in Scardapane et al. (2017), "Even if (the LASSO) has a non differentiable point in 0, in practice this rarely causes problems to standard first- order optimizers. In fact, it is common to simultaneously impose both weight-level sparsity with the $L_1$ norm, and weight minimization using the $L_2$ norm, resulting in the so-called elastic net penalization". This expectation is not being met in reality, and there is strong evidence of users struggling with the interpretation of their LASSO results in the PyTorch forums [4] [5] [6]. Given the wide variety of critical applications in which the LASSO finds use (Ghosh et al. (2021); Li et al. (2011); Li & Sillanpää (2012); Ogutu et al. (2012) to state a few), it is critical to understand the reliability of the NDGD solution for the LASSO problem.

---

[3]https://github.com/locuslab/edge-of-stability
[4]https://rb.gy/kh0qr
[5]https://rb.gy/2vsry
[6]https://rb.gy/6dfli

**NDGD solutions for the LASSO problem are not reliable:** We show that NDGD produces unreliable solutions even for the simple case of the LASSO penalized linear model. In particular, we show that increasing the LASSO penalty can result in a NDGD solution with a larger $L_1$ norm, completely defeating the purpose of the penalty. We first analytically demonstrate the unreliability for the simplest LASSO problem,

$$L_3(\beta) = L_1(\beta; 0, 0, \lambda_1, 0) = \lambda_1 ||\beta||_1, \tag{7}$$

with $\lambda_1 > 0$, and then demonstrate the same issues for the general LASSO problem through simulations. We begin by showing that for any non-zero learning rate, $\alpha$, the NDGD sequence for (7) will be non-sparse with probability 1.

**Proposition 2.** *Let $\{\beta_t\}_{t=0}^{\infty}$ denote the sequence generated by running NDGD on (7), with a constant learning rate, $\alpha$, and the entries in $\beta_0$ being uniformly sampled from $[-1, 1]$. Then $\beta_k$ will have all non-zero entries with probability 1 for all $k > 0$*

*Proof.* The NDGD iteration for (7) is described by the recursion

$$\beta_i[k] = \begin{cases} \beta_{i-1}[k] - \alpha\lambda_1 & \text{if } \beta_{i-1}[k] > 0 \\ \beta_{i-1}[k] + \alpha\lambda_1 & \text{if } \beta_{i-1}[k] < 0 \ . \\ 0 & \text{if } \beta_{i-1}[k] = 0 \end{cases} \tag{8}$$

From (8), it is clear that if the $k^{th}$ entry in the vector becomes zero during any iteration, then it stays zero for all subsequent iterations. Accordingly, suppose the $k^{th}$ entry in the NDGD sequence vector becomes 0 for the first time after $N$ iterations. Then we have

$$\beta_N[k] = \beta_0[k] + (N - 2m)\alpha\lambda_1 = 0, \tag{9}$$

where $m$ is the number of times the $k^{th}$ entry of the parameter exceeds 0 in the first $N$ iterations. (9) can only hold if $\beta_0[k]$, is an integer multiple of $\alpha\lambda_1$. Since the feasible values are countable, and the set of initializations is uncountable, the probability of the occurrence has measure 0. ∎

In the next proposition, we show that the sequence does not converge, but oscillates between two fixed points.

**Proposition 3.** *The sequence $\{\beta_t\}_{t=0}^{\infty}$ described in the previous proposition does not converge. Furthermore, there is an integer $N_0$ such that for every $N \geq N_0$, the sequence oscillates between two vectors : $\gamma_1$ and $\gamma_2$, where $||\gamma_1||_{\infty} < \alpha\lambda_1$, and $||\gamma_2||_{\infty} < \alpha\lambda_1$.*

*Proof.* We prove the results assuming $\beta_0[k] > 0$; the proof is similar when $\beta_0[k] < 0$. From Proposition 2, we know that for any $N > 0$, $\beta_N[k] \neq 0$ with probability 1. Therefore, (8) implies that starting from any $\beta_0[k] > 0$, the NDGD sequence will decrease monotonically till it reaches a value between 0 and $\alpha\lambda_1$. Accordingly, let $\beta_{n_k}[k] = \gamma_k$ for some $n_k \geq 0$, with $\gamma_k \in (0, \alpha\lambda_1)$. Since $\gamma_k - \alpha\lambda_1 < 0$, completing the recursion in (8) gives

$$\beta_{n_k+m}[k] = \begin{cases} \gamma_k - \alpha\lambda_1 & \text{if m is odd} \\ \gamma_k & \text{if m is even,} \end{cases} \tag{10}$$

The result follows by choosing $N_0 = \max(n_0, n_2 \dots n_{P-1})$. ∎

Equation (10) suggests the surprising result that with the same learning rate and initialization, a larger value of $\lambda_1$ can result in a NDGD solution with a larger $L_1$ norm, which completely defeats the purpose of the LASSO penalty. To better understand the issue, consider the 2D case, with the weights initialized as $\beta_0 = [0.5053, 0.5053]$. With $\lambda_1 = 1$ and $\alpha = 0.01$, Proposition 3 states that the NDGD sequence will oscillate between $[0.0053, 0.0053]$ and $[-0.0047, -0.0047]$, but with $\lambda_1 = 100$, and the same learning rate and starting point, the solution will oscillate between $[0.5053, 0.5053]$ and $[-0.4947, -0.4947]$. Thus, at the point when NDGD appears to converge, the solution with $\lambda_1 = 1$ will have an $L_1$ norm of 0.01, but solution with $\lambda_1 = 100$ case will have an $L_1$ norm of atleast 0.99. Furthermore, note that reducing the step-size does not resolve this paradox. If, in both cases, we decrease the learning rate to $\alpha = 0.001$ once the algorithm seems to have converged, then with $\lambda_1 = 1$, the NDGD sequence will oscillate between $[0.0003, 0.0003]$ and $[-0.0007, -0.0007]$. However, with

$\lambda_1 = 100$, the NDGD sequence will oscillate between $[0.0053, 0.0053]$ and $[-0.0947, -0.0947]$, and the paradox persists.

To demonstrate the same issue in the broader setting, consider the loss function for the general LASSO problem given by

$$L_4(\beta; \mathbf{W}, \mathbf{y}, \lambda_1) = \frac{1}{N}||y - W\beta||_2^2 + \lambda_1||\beta||_1, \tag{11}$$

where $\mathbf{W}$ is an arbitrary $20 \times 500$ dimensional data matrix, and $\mathbf{y}$ is an arbitrary $20 \times 1$ response, each of whose entries are sampled from the uniform distribution on $[-1, 1]$. We run NDGD on (11) twice with the same learning rate and initialization, once with $\lambda_1 = 0.01$, and once with $\lambda_1 = 10$. At the end of the two runs, the optimal solution with $\lambda_1 = 0.1$ has an $L_1$ norm of 1.62, and the optimal solution with $\lambda_1 = 10$ has an $L_1$ norm of 25.4 i.e., a 1000 fold increase in the value of $\lambda_1$ results in a more than 15 fold increase in the $L_1$ norm of the optimal solution!

The same pathology can be observed in more complex architectures like VGG16. To illustrate this, we initialize VGG16's weights using the pretrained values available in PyTorch. We then train the model using gradient descent on the same subset of 2000 images from CIFAR-10 twice, employing the $L_1$-penalized cross-entropy loss. In the first instance, we use $\lambda_1 = 0.001$, whereas in the second instance, we use $\lambda_1 = 0.1$. At the end of training we observe that the $L_1$ norm of the optimal solution is smaller in the first case (7312.07) than in the second (51707.23). We observe the same paradox with the widely used Adam (Kingma & Ba, 2014) optimizer.

**NDGD LASSO solutions and network pruning:**  Our intuition about the LASSO problem is that a larger LASSO penalty results in a smaller $L_1$ norm for the optimal solution (Hastie et al., 2015). This intuition is the basis for implementing $L_1$ penalization during the initial training phase of a neural network, preceding network pruning. Indeed, this is what NVIDIA recommends for the training of standard image classification networks like Retinanet and YOLOv4 [7]. The examples described above contradict this notion, demonstrating that employing the $L_1$ penalty alongside NDGDs can have an adverse effect, actually amplifying the $L_1$ norm of the optimal solution.

**Connections to the subgradient method:**  The counter-intuitive behavior seen in the examples above can be explained using the subgradient method. Since the loss function described in equation (7) satisfies the Lipschitz condition (see equation (6)) with $L = \lambda_1$, we have from Boyd et al. (2003) (Section 2 on the Constant Step Size) that

$$\lim_{k \to \infty} ||\beta_k||_1 < \alpha\lambda_1. \tag{12}$$

Therefore, for the same value of $\alpha$, the larger value of $\lambda_1$ will result in a larger error in estimation, which is precisely what we are seeing above.

**Diminishing v/s reducing step sizes:**  This illustration also highlights the contrast between the diminishing and reducing step size regimes outlined in the preliminaries. Under the diminishing step-size regime, $\alpha \to 0$ in the theoretical limit, and therefore, Proposition 2 guarantees that the optimal solution for (7) will be 0 for all $\lambda_1 > 0$. On the other hand, in the reducing step-size regime, $\alpha > 0$ at the end of training, and therefore, the error in estimation becomes a critical determinant of the optimal value.

**NDGD and the way forward with sparsity:**  While NDGDs do not inherently generate sparse solutions, iterative methods have been developed to produce sparse solutions for the $L_1$ regularized linear problem (see Table 1 in Bottou (2010)), and $L_1$ regularized neural networks in general (Ma et al., 2019; Siegel et al., 2020). The realization that NDGDs lack the capability to produce sparse solutions predates the widespread adoption of deep learning. As mentioned in Xiao (2009), "... stochastic gradient descent (SGD), has limited capability of exploiting problem structure in solving regularized learning problems". Despite this, a prevalent misconception persists in the deep learning literature, erroneously suggesting that such algorithms yield sparse solutions, and that the solutions generated by deep learning frameworks like PyTorch (Paszke et al., 2017) ought to be comparable to

---

[7]See the recommendation of the regularizer in the Training config section of `https://rb.gy/20y13m` and `https://rb.gy/i1sb6s`

those generated using traditional machines learning frameworks like glmnet Friedman et al. (2021) or scikit-learn Kramer & Kramer (2016). We point out that there exits a fundamental disparity between traditional machine learning frameworks like glmnet and deep learning frameworks like PyTorch. glmnet focuses on a narrow set of optimization problems, concealing implementation intricacies while guaranteeing optimality. Conversely, PyTorch provides greater flexibility in constructing models but lacks similar guarantees, placing the responsibility for solution quality on users. Attaining a solution at a non-differentiable point poses a challenging problem, underscoring the significance of exploring related studies and tools rather than opting for a simplistic implementation, particularly when precision is paramount.

**Connections to the Edge of Stability:** Proposition 3 shows that the NDGD sequence for the non-differentiable, Lipschitz continuous, convex loss function described in (7) will not diverge to $\infty$ for any finite value of $\alpha$. This is the special case of unstable convergence, a topic we cover in greater detail in the next section.

## 5 NDGD AND THE EDGE OF STABILITY

In recent years, there has been a growing interest on the topic of "unstable convergence". The literature on the subject Ahn et al. (2022a;b); Arora et al. (2022); Chen & Bruna (2022); Cohen et al. (2021); Li et al. (2022) is motivated by the finding that unlike convex quadratic forms, gradient descent on neural networks do not diverge to $\infty$ even when the learning rate is greater than $\alpha^* = 2/\eta$, where $\eta$ is the dominant eigenvalue of the loss function's Hessian. Instead, the loss function has been empirically demonstrated to converge "unstably" [8], reducing non-montonically on the long run. $\alpha = \alpha^*$ is referred to in the literature as the "Edge of Stability", since it separates regions of "stable convergence" ($\alpha < \alpha^*$) from regions of "unstable convergence" ($\alpha > \alpha^*$).

The derivation of the Edge of Stability condition assumes that the loss function, $f$, is $L - smooth$ (see equation (5)). Since the result has been demonstrated in most convex optimization settings Jacot et al. (2018); Li & Liang (2018), it is conjectured that the result ought to hold even in the case of non-differentiable neural networks. This conjecture is incorrect even for non-smooth, differentiable loss functions satisfying (6) as shown in the proposition below.

**Proposition 4.** *All convex non-smooth loss functions having bounded subgradients, or satisfying equation (6) will show unstable convergence.*

*Proof.* Note that any loss function satisfying (6) has subgradients bounded above by $L$. Additionally, as outlined in Section 1 of Section 1 of Boyd et al. (2003), the deductions derived from the subgradient method hold equivalently for differentiable functions. This equivalence arises from the fact that the only admissible value for the subgradient of a differentiable convex function at any given point is, indeed, its gradient. Accordingly, from Section 3 of Boyd et al. (2003), we have that $\lim_{k \to \infty} f(x_k) - f^* \leq \alpha L^2$, hence the result. ∎

In the previous section, we analytically derive the unstable convergence for the LASSO problem described in (7). An example of a differentiable non-smooth convex function satisfying Proposition 4 is the Huber loss function, which finds use in many state of the art object detection models (Girshick, 2015; Liu et al., 2016; Ren et al., 2015). The Huber loss function is non-smooth because it is once but not twice differentiable. For a regression problem involving an arbitrary $50 \times 1$ response vector, $\mathbf{y} = [y_1, y_2 \ldots y_N]^T$, and an arbitrary $50 \times 200$ data matrix, $\mathbf{Z}$, the Huber loss function is defined as

$$L_7(\beta) = \frac{1}{N} \sum l(i), \quad \text{where} \quad l(i) = \begin{cases} \frac{1}{2}(y_i - z_i^T \beta)^2 & \text{if } \left|(y_i - z_i^T \beta)\right| < 1 \\ \left(|y_i - z_i^T \beta| - \frac{1}{2}\right) & \text{otherwise} \end{cases}, \quad (13)$$

where $z_i$ is the $i^{th}$ column of $\mathbf{Z}$. Figure 3c shows that the NDGD sequence for (13) does not diverge toward $\infty$, even with a high learning rate, such as $\alpha = 10$, which aligns with our expectations. Furthermore, Figure 3b reveals an intriguing loss pattern when $\alpha = 1$: the loss initially increases before

---

[8]The most precise definition of (un)stableness we could was Definition 1.1 of Arora et al. (2022). Unfortunately, this definition requires the existence of a Hessian at all points between the current and next iterate, which is not true for ReLU networks. Accordingly, we use a generalization of their definition, assuming that unstable convergence means "non-divergence to $\infty$"

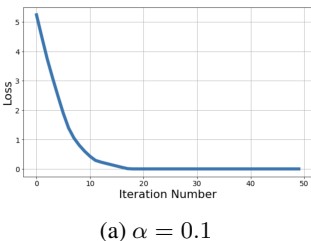 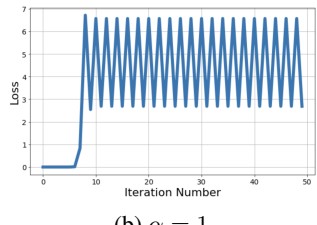 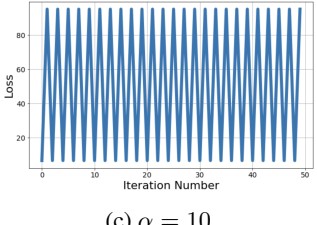

(a) $\alpha = 0.1$                     (b) $\alpha = 1$                     (c) $\alpha = 10$

Figure 3: Training curves for different values of $\alpha$, with the Lipschitz continuous Huber loss function. As the learning rate increases, the loss starts oscillating with larger frequencies, but never diverges to $\infty$. This is to be expected from Proposition 4, since the Huber loss function is convex, non-smooth and has bounded subgradients.

the oscillations become noticeable. This peculiar behavior has been documented in the literature (Ahn et al., 2022b; Arora et al., 2022), although a conclusive explanation for this phenomenon remains elusive at this time.

Proposition 4 and the examples that follow show that the common approach of extrapolating results developed for smooth loss functions to the non-smooth or non-differentiable case is problematic. In the next subsection, we demonstrate the problems with this approach by providing an example of an extrapolation which breaks down in the non-smooth case.

SUBQUADRATIC LOSSES AND UNSTABLE CONVERGENCE

In section 3.1 of Ma et al. (2022), the authors employ second-order finite differences to investigate the curvature of VGG16 and ResNet in the vicinity of the local minimum. They conclude that subquadratic growth of the loss function around this minimum point explains the edge of stability phenomenon.

To challenge the broad applicability of this statement, consider the Huber loss function depicted in Equation (13). This function is overparameterized and therefore possesses a global minimum of 0. Consequently, within the vicinity of this global minimum, the loss function exhibits quadratic growth. In such instances, Ma et al. (2022)'s result states that we will not see unstable convergence. However, contrary to their suggestions, Proposition 4 guarantees unstable convergence under these circumstances, as evidenced in Figure 3. The discrepency arises because the Huber loss function does not satisfy the condition of twice differentiability required by Definition 1 of their paper. This counter-example underscores the problems with extrapolating results developed results under strong assumptions (like $L - smoothness$ or twice differentiability) to the general non-differentiable case.

**The way forward:**   We have shown that our understanding of the Edge of Stability will be incomplete if we treat NDGDs akin to GDs. Furthermore, considering ReLU networks' non-differentiable minima, without further research into NDGD behavior around non-differentiable points our grasp of training dynamics lacks completeness. Consequently, we propose employing the convex non-differentiable single-layer neural network presented in Kumar (2023) as the toy problem for theory development. Studying this neural network offers several advantages: 1) its convex nature renders it more amenable to analysis compared to non-convex functions, 2) we possess analytical knowledge of the global optimum, and 3) the global optimum exists at a point of non-differentiability—the setup which is particularly relevant to ReLU networks.

## 6   CONCLUSION

In this paper, we highlight the impact of non-differentiability across three domains: convergence analysis, the LASSO problem, and the edge of stability. In each of the cases, we show that non-differentiability changes the dynamics of the training process, and introduces behaviors which cannot be explained using the theory of gradient descents in continuously differentiable functoins. The goal of our paper is identical to that of Bertoin et al. (2021) – to bring attention to non-differentiability which is often ignored in neural networks.

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
