# OpenReview forum: "Three ways that non-differentiability affects neural network training"
_ICLR.cc/2024/Conference — Submitted to ICLR 2024_

### Official Review · Reviewer_TRVW · 2023-10-14

**Soundness:** 2 fair
**Presentation:** 2 fair
**Contribution:** 1 poor
**Rating:** 3
**Confidence:** 4

**Summary:**

This paper discusses three ways that non-differentiability impacts network training: by affecting the convergence rate, causing unreliable LASSO solutions, and breaking results from edge-of-stability analysis.

**Strengths:**

1. The paper is clearly structured.
2. Non differentiability is an interesting and important component of modern neural networks.

**Weaknesses:**

1. The authors imply that small VGG16 network weights are close to the "domain of non-differentiability." This is flawed, as weights and activations are not the same -- if for example the bias of the linear layer is a very large constant vector, then we have differentiability even from small weights.
2. The authors generally frame their analysis as "practical networks do not satisfy theoretical ssumption (e.g., L-smooth loss functions), therefore they don't apply." This is a fundamental misunderstanding of the importance of theory. Theory tends to study a "toy" scenario, and derive results that are insightful even if they don't directly apply.
3. Generally, the paper consists of a few loosely connected ideas, and does not provide sufficient analysis for a conference at ICLR's level.
4. Presented theoretical results are very simple.

**Questions:**

1. The authors imply that edge of stability training doesn't hold for non-smooth networks. Does this not contradict with the cited Cohen paper, which uses standard networks with ReLU nonlinearities?

---

> ### Author Response · Authors · 2023-11-20
> **Thank you for your review**
>
> Thank you for your review. Please find our answers to your questions below
>
> 1) The image has been removed, and we have added a reference which shows that that ReLU networks do not have differentiable minima except in trivial cases.
>
> 2) A differentiable problem is not a good toy problem for a non-differentiable use case. That is the entire premise of our paper
>
> A couple of things in regard to the Cohen paper:
>
> 1) If anything, we agree with the conclusions made in that paper. In their discussions they mention that L-smoothness is not a good assumption to make when studying neural networks
>
> 2) Most of their paper  they use differentiable neural networks until the very end where they discuss ReLU networks under "experiments"
>
> 3) Note that in the gradient descent matching gradient flow argument, it held for all differentiable architectures but not some non-differentiable architectures. So there are differences in the non-differentiable case.

---

### Official Review · Reviewer_A5zk · 2023-10-27

**Soundness:** 2 fair
**Presentation:** 3 good
**Contribution:** 2 fair
**Rating:** 3
**Confidence:** 4

**Summary:**

This paper studies how non-differentiability affects the optimization trajectory of convex non-smooth minimization problems. The analysis comes from three perspectives: the convergence behavior, the algorithm’s ability to reach the optimal point, and the effect of non-differentiability on the phenomenon of the edge of stability. In general, the paper argues that non-differentiability introduces a different dynamic of optimization, and calls for a more fine-grained analysis to deal with non-differentiability.

**Strengths:**

1. The topic considered in this paper is interesting. Indeed, non-differentiability deviates from traditional regulatory conditions in convex optimization, and it is a mystery why blindly applying SGD on e.g. ReLU neural network can achieve a good performance.

2. The paper is well-written. The topics of the paper are decoupled nicely into three different sections, and each section is written clearly.

**Weaknesses:**

1. Objectives are too simple and not representative of neural networks. The paper considers a one-layer linear model with an added ReLU activation at the end, which is an extremely simple version of neural networks. It further assumes that the labels $\mathbf{y}$ contain only non-positive entries, which is almost never the case in real neural network training. In particular, the two conditions combined enforces the neural network to almost never fit all training labels.

2. The argument that the GD dynamic differs on differentiable and non-differentiable objectives is not strong. In particular, the contradiction is constructed by running experiments on the simple objectives discussed above, and no rigorous of the whole claim is provided.

3. The paper argues on Page 4 that ”it is consistently observed that the optimal parameter values ... non-differentiability for ReLU activations”. This argument is wrong since zeros of the parameters in the neural networks are not necessarily points at non-differentiability, which depends both on the neural network inputs and the bias.

4. The argument in the last paragraph on Page 4 is also not strong since Tibshirani 2015 only proves the upper bound on the number of steps for subgradient methods. Saying the upper bound of one algorithm is different than the upper bound of the other algorithm is not sufficient to say that the two algorithms have different convergent behavior since the slower convergence rate might be an artifact of proof.

5. The conclusion in Section 4 of the paper does not significantly deviate from previous viewpoints. In summary, the paper shows that the subgradient gradient method does not give sparse solutions for LASSO, which is a known fact. The paper also does not theoretically extend this argument to more complicated objectives such as neural networks.

6. In Section 5, the author argues that EoS have different behavior under non-differentiability, and uses Proposition 4 as a support. This is insufficient since the boundedness of the gradients is not related to either the progressive sharpening or the stabilization of the sharpness right above $2/\alpha$.

7. In the first paragraph of Page 8, the paper tries to use the setting in Eq. 12 to experimentally contradict Ma et al. (2022). However, Eq. 12 deviates a lot from the practical neural network setup and conclusions on Eq. 12 may not directly apply to neural networks.

**Questions:**

Please see "Weaknesses".

---

> ### Author Response · Authors · 2023-11-19
> **Thank you for your review**
>
> Thank you for your review. Please find answers to your questions attached:
>
> 1a. Our result are concerned with the structural properties of the NDGD sequence, and not with the specifics of the architecture, or how easy it is to train.
>
>
> 1b. While simulations on the general non-differentiable non-convex case do show similar signs of non-capturing, it is hard to draw conclusions from those simulations because we do not have apriori expectations in what we expect the GD sequence to do in this setup.The non-negative labels are chosen in our analysis because with these labels, the loss function is convex, and therefore, we have a better handle on what we can expect from the GD sequence.
>
> 1c. We do not agree with the claim that single layer neural networks with ReLU activations are simple and/or not relevant. Single layer networks are precisely what are being trained in most transfer learning problems as the "regression head" or the "classification head"
>
> 2) We do not agree with this claim either. An example of one type of sequence not satisfying the properties of another type of sequence is rigorous proof of the fact that the two types of sequence do not have identical properties.
>
> 3) Thank you for pointing this out. We have removed the figure, and cited [1], who show that barring trivial cases, the local minima of. ReLU networks will be non-differentiable
>
> 4) Thank you for pointing this out. The bound is tight. We have updated the reference in our paper to [2] to reflect this fact
>
> 5) We cite standard references in the field, highly cited papers, and concerns of unsure users on forums, all of whom expect the NDGD LASSO solutions to be sparse. Therefore the point is important to clarify.
>
> We have added results which shows that the paradoxical behavior seen with the penalized linear model is also seen with a penalized VGG16.
>
> 6) Our claims are primarily about unstable convergence, and we use the defn. of stableness provided in definition 1 of [3]. As mentioned in the above paper, definition 1 is preferable to the definition using sharpness because the loss can still oscillate in the latter case. A similar defn. is provided in [4] -  "More specifically, Cohen et al. (2021) observe that when we run GD to train a neural network, the condition (1.2) fails, but contrary to the common wisdom from convex optimization, the training loss still (non-monotonically) decreases in the long run....We call this phenomenon “unstable” convergence".
>
> 7) The Huber loss function used in our counter-example is used in many start of the are object detection models including Fast-RCNN, Faster - RCNN, and SSD. We have added references in our paper
>
>
> -------------
> [1] Laurent, Thomas, and James Brecht. "The multilinear structure of ReLU networks." International conference on machine learning. PMLR, 2018.
>
> [2] https://www.stat.cmu.edu/~ryantibs/convexopt-S15/scribes/07-sg-method-scribed.pdf
>
> [3] Arora, Sanjeev, Zhiyuan Li, and Abhishek Panigrahi. "Understanding gradient descent on the edge of stability in deep learning." International Conference on Machine Learning. PMLR, 2022.

---

> > ### Comment · Reviewer_A5zk · 2023-11-20
> > **Response to the Author Rebuttal**
> >
> > Thank you for your detailed response. I want to further share my opinion:
> >
> > 1. In my weakness #1, my main argument is that the objective considered in Eq. (3) is not representative of the class of neural network models, which the paper's title claims to be analyzing. In particular, the model is a linear model with an added ReLU at the very end. Although the author claims in 1b. that this is the model used in transfer learning, it must be noticed that it is very rare to append ReLU at the very end of a neural network. Moreover, the model in transfer learning is certainly a model easier to train than the whole network and thus is not representative as well. Lastly, it is almost never the case that we construct a neural network that can only output non-negative values, but with the goal to fit only negative labels. While the author argues that this is to make the loss convex, I believe that this consideration is an oversimplification of the neural network case since [1] shows that neural network training is almost never convex.
> >
> > 2. My argument in my weakness #2 is that, since the author is analyzing a simplified model of neural networks, the "proof" given by the author should be rigorously mathematical. In the paper, the author shows the property of "one type of sequence" mathematically but shows the property of "another type of sequence" experimentally, and tries to compare the two to draw a conclusion. In my opinion, this analysis is still based on empirical observation and is not the full mathematical characterization we are hoping to see in this simple model.
> >
> > 3. Thank you for your response to my weakness #3. Right now I am convinced that this argument is correct.
> >
> > 4. It does not seem that the provided material given in the response shows that the proved bound is tight for the sub-gradient method. Usually, when one tries to show that a bound is tight, one will try to show that the upper bound matches the lower bound (possibly within a constant factor of difference). However, no such bound is given in the shared material. Moreover, I believe that to support an argument in a formal paper, one should avoid citing unpublished lecture notes, especially those scribed by the students.
> >
> > 5. In my weakness #5, I am saying that the results in Section 4 of the paper do not show improvement/changes/differences from previous results. Please correct me if I am wrong, it seems to me that your analysis is simply repeating what has been done and is not novel.
> >
> > 6. The paper by Arora et al. uses the definition to characterize smooth objectives. In particular, you could see in their Assumption 4.1 that the objective needs to be $C^4$. The same definition does not transfer to non-smooth objectives since the second-order derivative is undefined. I believe that the author should not use an ambiguous definition like this to make his main argument in the paper. I would suggest the author to maybe consider an extension of this definition.
> >
> > 7. While Huber loss is used in Fast-RCNN, Faster-RCNN, and SSD, the paper by Ma et al. considers models including VGG and ResNet. That is, the scenarios considered by the author and Ma et al. are non-overlapping and therefore do not stand as a case to conduct a comparison. Moreover, the model passed in the Huber loss is again a linear model and is thus not representative of very deep models like VGG and ResNet as well.
> >
> > [1] Chaoyue Liu, Libin Zhu, and Mikhail Belkin. "Loss landscapes and optimization in overparameterized non-linear systems and neural networks." 2020.

---

> ### Author Response · Authors · 2023-11-21
> **Response to concerns**
>
> Thank you for your follow up. Please find our responses below:
>
> 1) Your arguments reinforce our results. Your assessment is accurate in that we are only considering convex neural network, though neural networks are generally non-convex. Therefore, we are showing that even for the best case scenario of an oversimplified convex neural network, the hypothesis that NDGDs and GDs have identical properties is incorrect.
>
> 2) Using simulations to demonstrate counter-examples is a standard practice [1][2]. As mentioned in page 9 of [2], 47% of the studies they analyzed use graphical methods to demonstrate counter-examples. This is also the approach we use in our paper.
>
> 3) Thank you!
>
> 4) Section 7.1.3 of Tibshirani's lecture notes states that "Thus for an optimal solution, we need $O(1/\epsilon^{2})$
> iterations. Although the derivation shown here does not make this a tight bound, this bound turns out to be tight and
> this convergence rate is the best you can do for the subgradient method.". His notes are using the standard definitions of tightness. The same point is alluded to in Boyd's notes, where the first paragraph states "The subgradient method is far slower than Newton’s method, but is much simpler and can be applied to a far wider variety of problems".  The reason we chose to refer to lecture notes rather than texts is because 1) the former often times give a concise review of the key results we are interested in and 2) the latter is often behind pay walls. We point out that Dr. Tibshirani is a reputed expert in the field, and it is typical for course notes to be reviewed by the instructors before they go live. Boyd's lecture notes on the subgradient method have been cited more than 1200 times.
>
> 5) As we previously mentioned, we believe that the fact that LASSO solutions with NDGD produce non-sparse solutions is non-trivial; many standard references in deep learning suggest that an L1 penalized neural network will produce sparse solution. Additionally, we show that increasing the LASSO penalty can result in an NDGD optimal solution which has a larger L1 norm, violating our intuition for the problem. We show this for the case of linear models and VGG via simulation. We are not aware of anyone else who has demonstrated this result. We also show that this result has practical applications -- NVIDIA recommends using the L1 norm during training because they are using this intuition about the LASSO from the optimization/statistics literature, and our results show that this approach can backfire because L1 penalization behaves counter-intuitively.
>
> 6) We chose to use Definition 1.1 in Arora's paper to define (un)stableness precisely becaise is not ambiguous, and does not rely on the smoothness of the objective function. We do not use assumption 4.1 for the precise reason you mention - it is developed for smooth functions, and need not apply to the non-smooth case.
>
> 7) When you say Huber loss is not a linear model, we are assuming this is a typo, and you meant "regression model", or are we misunderstanding your point?  There are two issues which appear to be getting intertwined here. 1) We state that the results of Ma et al. do not work for the general case using the counter-example of the Huber loss function 2) We state that their results using VGG16 lacks interpretability because they perform second order differences around points of non-differentiability. Ma et al. make a claim about the properties of functions which show unstable convergence, and they use simulations to justify their results outside the assumptions in which their result is derived. Our counter-example shows that the approach of extrapolating results outside the domain in which they are derived need not work. The depth of the neural network is not relevant to the problem. Note that a deep network with the Huber loss will also show unstable convergence, because the subgradient is bounded.
>
> ---
>
> [1] Brayton, Alan Mishchenko Niklas Een Robert. "A Toolbox for Counter-Example Analysis and Optimization."
>
> [2] Kaleeswaran, Arut Prakash, et al. "A systematic literature review on counterexample explanation." Information and Software Technology 145 (2022): 106800.

---

> > ### Comment · Reviewer_A5zk · 2023-11-21
> > **Further Response**
> >
> > For each bullet point, my response is below:
> >
> > 1. I have a different opinion. I believe that the significance of this theory lies in that it explains (or sheds light on) the behavior of neural network training that we use in practice. In my perspective, it is less meaningful to claim that "not all neural network training satisfies property A" by constructing a counter-example that is never used in practice, since ultimately what we care about are the majority of neural networks that have real applicability. I believe that the convex neural network you present deviates quite a lot from what is used in practice.
> >
> > 2. I think [1] and [2] in your response are papers that consider formal verification, which is highly irrelevant to the topic considered in this conference. Moreover, I believe that the "graphic counterexample" stated in [2] means state machine or block diagram, which are formal ways to prove properties for algorithms. I don't think the plots used in your proof fall into this category. Therefore, I still believe that the proof lacks mathematical rigorousness.
> >
> > 3.
> >
> > 4. I do agree that both Dr. Tibshirani and Dr. Boyd are well-respected scientists in the field, and I understand that it is hard to get access to paid materials. However, to guarantee the rigorousness and correctness of the papers published in ICLR, I still believe that in order to argue something like "the upper bound matches a certain lower bound" you need to cite actual papers, in particular, formal theorems that show such property rather than discussions in paragraphs.
> >
> > 5. There might be a difference between NVIDIA's recommendation of "using LASSO" and "using LASSO with gradient descent". If "using LASSO with gradient descent" are known to have a problem in the field and it is advised that other optimizers are used to minimize LASSO objectives to achieve a sparse solution, then the author's claim would be significant only if the same property is shown for other optimizers (like adam and adagrad) as well. In particular, for a more precise discussion, I believe that the author needs to cite more resources that discuss this behavior.
> >
> > 6. The unstableness defined in Arora et al. relies on the largest eigenvalue of the Hessian. At non-smooth points, the Hessian does not exist so the definition does not apply. Therefore, the definition does not seem to directly apply to the scenario that the author is considering.
> >
> > 7. By model I meant predictor: namely when fed in example $z_i$, it produces $z_i^\top\beta$ as shown in Eq. (13) of the paper. This is a linear model. Reading the author's response in this bullet point, I think I understand more about the author's intent. However, I am now confused about Proposition 4. Isn't the proof of Proposition 4 showing the convergence of functions with bounded gradient or smoothness, which is precisely contrary to the unstable convergence claimed in the proposition? Moreover,  another issue is that (similar to what I mentioned in 1) if the counterexample is not what we used in practice, then I believe that is not so significant. It is the fact that the model is not deep which makes the counterexample to deviate from what is used in practice.

---

> ### Author Response · Authors · 2023-11-21
> **Further response**
>
> $\textbf{Answers to questions 1 and 2}:$ There are several points raised in this question, and we address each of them below
>
> Proof rigor: Our proofs are rigorous. In our rebuttal, we chose papers on formal verifications, because they are closely tied to the the ideas of how counter-examples are constructed; we apologize for not making the point clearer in our previous response. To make the point clearer, we note that such counterexamples are available in the optimization [1] and NLP [2] literature.
>
> Data patterns are not reflective of what we do in training: A critical design choice in PyTorch and other such deep learning frameworks is that the optimizer (which is what we study in this paper) is "data agnostic".  To point this out, we note that in the optimizer described in https://pytorch.org/tutorials/beginner/basics/optimization_tutorial.html#optimization-loop, we only pass in the model parameters and the optimization configuration. The optimizer does not know (or care) about the pattern of data being passed into the model.
>
> The general non-convex case: For the general  case, our paper has results from simulations to show that convergence in differentiable neural networks is faster than non-differentiable neural networks, a result which is not known before.
>
> $\textbf{Answers to question 4}:$ In our preliminaries, we mention that the books by Shor and Bertsekas are standard references on the subject, and the notes are an easy reference of the key results. All of the results on the tightness of the bounds and the convergence properties of the subgradient methods can be found in the above books, which the interested user can look up.
>
> $\textbf{Answers to question 5}:$ All of the algorithms you describe in your paper are derivatives of gradient descent [3]. Theoretical validations that the results not holding for Adam are undoubtedly important, but those subjects are out of scope of our current work. You correctly point out that users will be interested to know if our results hold with Adam too, and we have added results from simulations which show that they hold for Adam too. We have also added references for the intuition on the LASSO problem
>
> $\textbf{Answers to question 6:}$ Thank you for raising this point! It is worth clarifying. You are correct in stating that Arora's definition requires the existence of the second derivative for every point between the current and next iterate. This is precisely the reason why our analyses uses the definition of "non divergence to infinity", which does not rely on the differentiability of the function at all. This idea was not precise enough in the previous iteration; we have updated it.
>
> $\textbf{Answer to question 7:}$ Thank you for pointing this out! Proposition 4 in Boyd's results holds for any Lipschitz continuous, convex function -- differentiable or otherwise as seen in the proof in section 1 of their notes. When the function is differentiable, the only choice of the sub-gradient is the gradient. This is a point worth mentioning, which we have added to our proof.
>
> As for the concerns about not being used in practice, we have two responses. First, our purpose here is to point out the perils of using a mathematical result outside the assumptions in which it is derived. Second, as you acknowledge in your previous response, the loss function is used in practice -- in Fast-RCNN, Faster- RCNN and SSD. We are not clear on the expectation here.
>
> -----
>
> [1] Diaby, Moustapha, Mark H. Karwan, and Lei Sun. "On modelling hard combinatorial optimisation problems as linear programs: refutations of the'unconditional impossibility'claims." International Journal of Operational Research 40.2 (2021): 261-278.
>
> [2] Isradisaikul, Chinawat, and Andrew C. Myers. "Finding counterexamples from parsing conflicts." ACM SIGPLAN Notices 50.6 (2015): 555-564.
>
> [3] Netrapalli, Praneeth. "Stochastic gradient descent and its variants in machine learning." Journal of the Indian Institute of Science 99.2 (2019): 201-213.

---

### Official Review · Reviewer_SuQn · 2023-10-29

**Soundness:** 1 poor
**Presentation:** 1 poor
**Contribution:** 1 poor
**Rating:** 3
**Confidence:** 4

**Summary:**

This paper tries to investigate how non-differentiability affects machine learning algorithms. Several issues are raised when non-differentiability is present, including convergence rate under smooth conditions, convergence for LASSO, and reason for the edge of stability. Dozens of existing works are used to explain these issues.

**Strengths:**

This paper studies non-differentiability in deep learning, which is an important issue.

**Weaknesses:**

1. I am really confused about what the algorithm "NGD" is. The authors say it is "Non-differentiable gradient descents" without providing any concrete definition (how is $\tilde{\nabla}$ defined? Is it subgradient? If it depends on the problem, what is the concrete definition of it in Figure 1?). This makes it hard to evaluate this paper, as I do not know if it is the version of gradient applied to deep learning practice. Also, I would like to remind the authors that "NGD" is usually referred to as "Natural Gradient Descent", and I suggest the authors change the name.

2. The presentation of this paper needs to be polished. As an example, it is quite weird that the left subfigure of Figure 1 is not in a log scale, since there are only two or three points in the figure much above 0. Also, the subfigure even appears to be a screenshot with inconsistent color with the background. Also, on page 5, I do not think it is a proper way to use discussions in a forum to support claims.

3. Is Prop 1 correct? I do not check the reference, but at first glance, this appears to be strange, as intuitively it is possible that at $\theta_0$ the gradient is quite large and the iterations jump outside the unit ball.

4. This paper extensively cites conclusions from other papers without any further explanation. For example,
> Since the loss function described in equation (6) satisfies the Lipschitz condition (see equation (5)) with L = λ1, we have from Boyd et al. (2003) (Section 2 on the Constant Step Size) that...

5. Many connections are drawn based on imprecise observations/statements. For example, in the texts below,
> we know from Figure 2 that the optimal weights lies close to points of non-differentiability
How "close" is needed to make the Taylor expansion invalid?



**Typos**

1. The first line under Eq. (3): y is a $N\times 1$ ...?

2. Proof of Proposition 1: missing "."

**Questions:**

See the Weaknesses above.

---

> ### Author Response · Authors · 2023-11-19
> **Thank you for your reviews:**
>
> Please find our responses and changes we have made below:
>
> 1) We have renamed the sequence to NDGD instead of NGD. We explain the symbols after equation 2 where we first mention it -- it is the gradient if the gradient at the point exist, and a heuristic measure otherwise.
>
> 2) We have fixed the image. We believe that the claims in the forum are valid to cite because they have no bug in their code, and are raising a valid question given that multiple standard sources on the subject (cited in our paper) suggest that the NDGD solution ought to be sparse.
>
> 3) The result is correct. See page 15 of https://people.eecs.berkeley.edu/~wainwrig/ee227a/Bertsekas_1.2.pdf
>
> 4) We have added a subsection on the subgradient method in the preliminaries, and explain why Boyd's notes are cited in multiple locations in our paper.
>
> 5) We realize that our histogram was causing confusions. We have removed it and added a reference to [1] who show that ReLU networks have non-differentiable minima except in trivial cases. We have changed the statement to "around points of non-differentiability", to remove imprecise terms like "close
>
> 6) Both typos have been fixed. Thank you for pointing them out!
>
>
>
> ======
>
> [1]  Laurent, Thomas, and James Brecht. "The multilinear structure of ReLU networks." International conference on machine learning. PMLR, 2018.

---

### Official Review · Reviewer_AGDk · 2023-10-30

**Soundness:** 3 good
**Presentation:** 3 good
**Contribution:** 1 poor
**Rating:** 3
**Confidence:** 3

**Summary:**

This paper focuses on ways that non-differentiability may affect the training of neural networks. Specifically, the paper considers one layer fully-connected networks with ReLU activation functions, lasso and ridge regression. The paper shows three settings under which the training of neural network may have undesirable behavior. Specifically, the convergence rate of non-asymptotic NGD (non-gradient descent, as opposed to GD, gradient descnet), the lasso problem and the edge-of-stability problem.

**Strengths:**

The paper is mostly well-written, both in terms of language and, flow and literature review. It is also sound. The paper may motivate research towards the training of non-differentiable neural networks.

**Weaknesses:**

The main motivation for the work is that the theoretical results for gradient-descent methods on smooth functions may not hold in the case of non-smooth functions (second paragraph of the introduction), even though GD methods are used in non-differentiable cases in practice. This motivation, while true, is not too strong. As the authors acknowledge, it is a common violation in practical machine learning, and its success is undeniable. Therefore, I think the work would benefit significantly from having more significant, large-scale examples, where we see the non-differentiability as a clear problem. Considering in the examples non-asymptotic and constant learning rate settings is also distracting, as this way we can not distinguish if it the asymptotic and diminishing learning rate settings would hold the same problems.

The three specific arguments for drawbacks of non-differentiability are too disconnected and the paper also does not go too deep into each of these problems, and the results are too superficial and lacking a significant novelty. However, I do think each of the problems could be interested on its own, and could deserve more exploration from the authors. Specifically, while the authors point out problems, they do not investigate the impact of obvious solutions: if the problem is the constant learning rate, we can just make it diminishing; if the problem is non-differentiability, for instance from the ReLU layer or the lasso penalty, we can smoothen the function at points.

Finally, even though the work calls attention for three problems within the training of non-differential neural networks, investigating specific solutions, or at least directions for solving the problems raised, would make the paper much stronger. It could also allow for more specific and interesting conclusions than the one presented (which is that attention to the problem is required).

Minor and detailed comments:
- in the abstract, "under-estimate" is ambiguous;
- in the abstract, "unreliable" is also not clear;
- in the preliminaries, the learning rate $\alpha$ should be defined when it appears. the vectors $z_i$ are also not defined, even though we can understad their dimensions for the dimensions of the matrix $\mathbf{Z}$. The response vector is defined bold but appears un-bolded in the expression. Its dimension should also be $N \times 1$, not $P \times 1$.
- in the preliminaries, if it is argued that the differences between subgradient and standard gradient play a key role in the analysis, subgradient must be introduced, defined and discussed.
- the results in section 3 may be too unsurprising, as the authors acknowledge after discussing the experiments whose results appear in figure 1.
- the results in section 4 are more interesting. Nevertheless, the constant learning rate plays a key role in the contributions, and even in the differentiable case, non-convergence can happen with non-diminishing learning rates.
- the results in section 5 are too disconnected from the previous and makes the reading not transition smoothly.
- in section 5, the result cited in the proof holds for decreasing learning rates.

**Questions:**

- why is the loss function in equation (12) not differentiable ? or is it differentiable once but not twice ?
- where do the authors "demonstrate that the widely-used assumption of L-smoothness tends to significantly overestimate convergence rates"? Is this not shown in Tibshirani (2015)?
- in the second paragraph of section 3, why is the reference for saying "the learning rate (is) reduced three of four times during training" ?
- in the third paragraph of section 3, "optimal" is in the local or global sense ?
- do the authors believe the solution for the problem raised should be more directed to the function (smoothen the function to differentiability) or the algorithm (develop specific algorithms for non-differential functions) ? Can this discussion be added ?
- in section 5, the result cited in the proof holds for decreasing learning rates. is that not contradictory to the constant learning rate setting considered ?

---

> ### Author Response · Authors · 2023-11-20
> **Thank you for your insightful review**
>
> We agree with a lot of the points that you make, and have made the following changes to address them:
>
> 1) We have re-written the introduction emphasizing why the problem is important.
>
> 2) We rewrote the preliminaries to address your question as to why the diminishing step-size is not being used. Essentially, we explain why every real world training is a constant step-size problem.
>
> 3) In our section on convergence analysis, we added simulations showing that for deep fully connected and convolutional neural networks, differentiable activations result in faster convergence than non-differentiable activations. We have also added the "way forward" where we talk about recent work on differentiable neural networks.
>
> 4) In our section on the LASSO, we have added results from more simulations, and added the way forward with sparsity in neural networks. We explain the connections to reducing step-sizes. We have also added the bridge to the section on the "Edge of Stability"; the toy problem we discuss in the LASSO is an analytical derivation of the edge of stability criteria.
>
> 5) For the case of the "Edge of Stability", the Huber loss function is indeed differentiable, but non-smooth! Thank you for catching this sizable bug. Added the way forward and suggest a prototype toy problem for the study of the Edge of Stability problem.
>
> 6) The result in section 5 holds with the constant learning rate. The last line on the constant step size in his notes: "We find that $f(x^} - f^{*}<=G^{2}\alpha ...".
>
> 7) Ambiguous terms have been removed.
>
> 8) We do not believe that the fact that NDGD and GD dynamics are not identical is an unsurprising point. We have had multiple reviews question the validity of our result

---

### Meta-Review · Area_Chair_wFuq · 2023-12-10

**Metareview:**

This paper discusses three ways that non-differentiability impacts network training: by affecting the convergence rate, causing unreliable LASSO solutions, and breaking results from edge-of-stability analysis.

There is consensus among the reviewers that the paper is hand-waving. It consists of a few loosely connected ideas, and does not provide sufficient analysis and rigor.

**Justification For Why Not Higher Score:**

There is consensus among the reviewers that the paper is hand-waving. It consists of a few loosely connected ideas, and does not provide sufficient analysis and rigor.

**Justification For Why Not Lower Score:**

N/A

---

### Decision · Program_Chairs · 2024-01-16

Reject